# COVID-19 and Bangladeshi health professionals: Infection status, vaccination and its immediate health consequences

**Bilkis Banu**[☯]*, **Nasrin Akter**[☯], **Sujana Haque Chowdhury**[‡], **Kazi Rakibul Islam**[‡], **Md. Tanzeerul Islam**[‡], **Muhammad Zahangir**[‡], **Shah Monir Hossain**, **Sarder Mahmud Hossain**

Department of Public Health, Northern University Bangladesh, Dhaka, Bangladesh

☯ These authors contributed equally to this work.
‡ SHC, KRI, MTI and MZ also contributed equally to this work.
* bilkisbanu80@gmail.com

**Data Availability Statement:** All relevant data are within the paper and its Supporting information files.

## Abstract

Healthcare professionals play a pivotal role in protecting and saving the lives of general people. As health workers are more likely to be infected with COVID-19, it is inevitable to safeguard them through vaccination in advance to continue healthcare services. Hence the study aimed to explore the infection and vaccination status along with immediate health consequences among these frontiers. This was a cross-sectional, web-delivered study conducted among the 300 healthcare frontiers working at COVID-19 dedicated hospitals in eight divisions of Bangladesh. The study questionnaire encompasses infection, vaccination status with dose information, and demographical and organizational information among the respondents. A multivariate logistic regression model and Chi-square test was used for the analytical exploration. Adjusted and Unadjusted Odds Ratio with a 95% confidence interval was calculated for the specified setting indicators. The study revealed that 49% of all respondents tested positive whereas 98% of them were found vaccinated of which mostly (52.3%) had their 2nddoses and 68.7% faced immediate health consequences for having the vaccination. As predictor for COVID-19 infection status, young and senior adult group (30–39 years: AOR = 2.01/0.03; 95% CI: 1.08–3.76; >50 years: AOR = 4.36/0.01; 95% CI: 1.65–11.55) and respondents who received Sinopharm as their vaccine found to have more significant positive infection history. The predictors regarding experiencing immediate health effects after vaccination, surprisingly female (AOR = 3.31/0.01; 95% CI: 1.82–6.04) health professionals of the capital city (AOR = 1.91/0.03; 95% CI: 1.06–3.46) were observed to have health consequences on vaccination. As the older female group (>50 years) in the nursing profession was found more infected with COVID-19 and a significant number of health professionals especially the age group (30–39 years) in the nursing profession experienced immediate health effects of COVID-19 vaccination, implementation of specific strategies and policies are needed to ensure the safety precaution and effective vaccination among the health professionals of Bangladesh.

**Funding:** The author(s) received no specific funding for this work.

**Competing interests:** The authors have declared that no competing interests exist.

## Introduction

COVID-19 pandemic is a name of the recent curse tremendously witnessed and suffered by all appalled populations throughout the globe. Up to 16 May 2022, the pandemic had caused more than 521 million cases & 6.2 million deaths, making it one of the deadliest in history [1]. Effective preventive strategies have been suggested to prevent transmission of COVID-19 such as wearing a face mask, repeated hand washing, travel ban, social distancing, etc. [2]. In addition, as an effective and sustainable solution to combat, the pandemic worldwide leaders turned towards vaccination with the successful development, evaluation, and production of multiple vaccines [3].

During the pandemic, healthcare professionals require direct contact with COVID-19 patients to save the infected patients in the healthcare setting [4]. The World Health Organization (WHO) and the Centers for Disease Control and Prevention (CDC) have identified health care workers (HCWs) as a population with a significantly elevated risk of being infected with the deadly disease. Thus, as frontline fighters like other significant professionals the healthcare professionals are significantly vulnerable during this pandemic owing to their commitment to mitigating the disease [5–8]. Bangladesh has a long experience of health workforce crisis with an absolute shortage of health worker. There are 7.7 legally qualified registered health care providers (HCP), such as doctors, nurses, and dentists, per 10,000 people, making up just about 5% of the entire health workforce [9]. The death rate is the highest among doctors affecting coronavirus in Bangladesh. For instance, as of October 15, 2020, there were around 4,797 COVID-19 cases among doctors and nurses, with more than 100 deaths of physicians in Bangladesh [10]. A study of coronavirus infection observed 10.79% among health care workers in a COVID dedicated tertiary care hospital in Dhaka, Bangladesh [11]. Another study among HCWs throughout Bangladesh from an academic platform revealed that 41% of healthcare professionals were diagnosed with a positive infection in their healthcare setting [12]. Moreover, such public health emergency affects the mental health of healthcare workers, including professional stress, fear of infection, and feeling helpless [13]. Thus, this miserable scenario recommends the rapid and prioritized vaccination of HCWs against COVID-19 to protect them as well as the whole population [14].

During health emergencies, especially with a pandemic of an infectious disease like COVID-19, contamination spreads quickly around the world and causes millions of deaths. In such situations, besides individual & social preventive measures; vaccines are the most powerful tool to save billions of lives. One of the biggest success stories in the history of public health is the development of vaccines; in 1977, the smallpox virus was successfully eradicated, and the wild poliovirus is now about to exterminate [15].

Like other countries, Bangladesh also facilitated a mass vaccination program in 2021 to protect the people from the COVID-19 pandemic. The program was figured out according to the prioritizations and effective implementation plans for the COVID-19 frontiers [16]. Though initially, the worldwide acceptance of vaccination was not satisfactory willingness towards vaccination was found positive among the Bangladeshi population [17]. However, usually approved vaccines had some immediate consequences that people witnessed. According to public health experts, all these side effects are immediate temporary reactions and these are signs that the response of our immune system over such antigen and being ready to protect people from COVID-19 [18, 19]. In Bangladesh, the government systemically monitors the adverse events that might occur following covid-19 National Vaccine Deployment Plan (NVDA) [20].

As outlined in the Strategic Advisory Group of Experts (SAGE) roadmap, vaccination should be prioritized for high-risk groups such as health workers, older adults and immune-

compromised populations, refugees, and migrants [21]. Thus, it was visualized that almost all health workforces would be under vaccination on a priority basis, as they are frontline fighters during this pandemic. Our previous study among healthcare professionals revealed that a good number of healthcare workers (18.3%) were non-vaccinated till August 2021. Reasons for non-vaccination were found as registration issues (52.70%), misconceptions regarding vaccination (29.10%), and health-related issues (18.20%). The study also showed a higher infection rate (41%) among the HCWs as they need to be in close contact with the positive cases during handling the cases and curing them. While all of them were not covered within the vaccination program [12]. However, a study on vaccination among HCWs was conducted only based on academic platforms. As HCWs are the frontline fighters of this current pandemic we need to have a detailed snapshot regarding such issues from all HCWs populations throughout the country.

As there are fewer data available regarding this issue our study was intended to explore a new dimension of nationwide updated COVID-19 infection and vaccination status with the immediate health consequences of the vaccine among the HCWs of Bangladesh with the actual representative sample from the COVID-19 dedicated hospitals all over the country. The findings of the study may an effective initiative for the health care professionals of Bangladesh regarding their health safety policies. It will also influence the government to undertake a sustainable program with the aim to handle such sudden health-related catastrophes like COVID-19 and combat the situation successfully.

## Methods

### Study design & setting

This was a cross-sectional study based on a descriptive approach followed quantitative design. Structured data were collected in this study from January to February 2022 for extracting information on infection status, vaccination, and its immediate health consequences on COVID-19 among Bangladeshi health professionals. Respondents were selected from 16 COVID dedicated hospitals (01 Government and 01 private from each division) under 8 divisions of Bangladesh. Selected hospitals exclusively maintained all the measures of infection, prevention and management followed national guidelines [22].

### Study participants, sample size and sampling

This study included a total of 300 respondents from eight divisions of Bangladesh. The respondents signified as active healthcare professionals such as physicians, nurses, and allied health personnel i.e. nutritionists, physiotherapists, laboratory technologists, etc. serving in different public and private healthcare organizations across the country. Healthcare providers of COVID-19 dedicated hospitals of eight divisions (Dhaka, Chittagong, Rajshahi, Khulna, Rangpur, Mymensingh, Sylhet, and Barisal) of Bangladesh were considered as the study population. Hospitals that served the greatest number of COVID-19 patients during the pandemic were taken into account as study places.

This study considered a 300-sample size. Initially, it was assumed that a potential standard sample size of 372 would be taken by using the formula "n = '$Z^2pq/d^2$'" where Z (standard normal deviate) was considered as 1.96; p (proportion of infected healthcare professionals) was considered as 0.41 [12] and margin of error was considered as 0.05. However, the final sample size was directed to 316 by the declined respondent rate of 15% according to their response to the self-administered data collection instrument. After data cleaning and initial management, the final samples were fixed at 300.

Considering the hierarchy of infection rates among eight divisions of Bangladesh [23], 50% (150 respondents) of the study subjects were recruited in the study from the capital of Bangladesh i.e. Dhaka division, and the remaining 50% (150 respondents) were enrolled from outside capital i.e. other seven divisions. One government and one private hospital were selected from the COVID dedicated hospital list of each division. Therefore, two hospitals selected from each division directed a total of 16 hospital selections from the country. Respondents were selected randomly from the health care provider list collected from the hospitals who were physically fit and had a willingness to participate in this study.

## Data collection

A structured and anonymous online questionnaire was used to gather data using a self-administered method. The questionnaire was developed using google form and response was limited to one against each google sign-in for avoiding duplicity of the responses including checking their personal information. A physical and paper-based questionnaire was avoided considering the spread of theCOVID-19 pandemic situation and for the speedy collection of this pivotal information. Respondents were recruited on January 2022 in this study and accessed through emails and/or WhatsApp and/or Facebook Messenger concurrently. The weblink of the online survey was 'https://docs.google.com/forms/d/e/1FAIpQLScSS5Q7dmgAR-rgyX0L1pexGYk Qi-kFyZaK4u_7W_R8ZpJVnQ/viewform?usp=sf_link' which took only 5 to 6 minutes by the respondents to complete. All authors had access to the collection and preserving of participants' information during or after data collection. The online web-based survey was administered in the English language with the utmost support of the hospital authority.

## Ethical considerations

This study was approved by the Ethical Review Committee of the Department of Public Health of Northern University Bangladesh (NUB/DPH/EC/2022/12-a) and conformed to the Declaration of Helsinki. Participation of the respondents was anonymous and voluntary. Informed consent was sought in a written format from the respondents at the beginning of the survey and participants could withdraw from the survey at any time.

## Questionnaire design

The online questionnaire was developed using Google Forms. The questionnaire was pre-validated by two independent reviewers following the variables of our previous study [11]. The pre-test was done to finalize the questionnaire among 10 respondents reported in the COVID dedicated hospital of Dhaka city except for the selected hospitals of this study. The quality of the questionnaire addressed the responses of the pre-test. The questionnaire comprised of several segments: (i) Identification of COVID-19 infection status who stated that they had COVID-19 confirmatory test by RT-PCR [12]; (ii) Reveal of COVID-19 vaccination status including stages of doses (1st dose/ 2nd dose/ 3rd dose) with the brand name; (iii) Demography and organizational information of the healthcare professionals: age, gender, profession, geographical location, occupation, organization type.

## Data analysis

Collected data was checked and analyzed employing the Statistical Package for the Social Sciences (SPSS) software. Study characteristics were subjected to descriptive statistics (frequency and proportions) to summarize the obtained data. To categorize the data of age, the cut-off value was decided according to previous relevant published articles [12]. A multivariable

logistic regression analysis was performed followed by a modeling procedure considering the backward elimination process, including pre-specified confounders i.e. age, gender, profession, geographical location, occupation, and organization type. Adjusted Odds Ratios with95% confidence intervals with respect to COVID-19 infection (test positive or test negative) and vaccination status (vaccinated or non-vaccinated) were calculated for the specified exposures.

## Results

### Participant's characteristics

A total of 300 respondents were included in this study where the response rate was 84.9% (316/372). Demographic characteristics reflected that 69% were female and nearly half of the respondents (38.3%, n = 115/300) belonged to the age group of 30–39 years with (mean±SD), (36.13±9.13). As HCWs, the highest (44.7%, n = 134/300) respondents were nurses. For nationwide data coverage 50% of total data were collected from the capital of Bangladesh: Dhaka district (n = 150) and 50% from outside the capital (n = 150). In addition, more than half of the study subjects (70.3%, n = 211/300) were found as employers of public health care organizations, while 29.7% were from private organizations. Furthermore, the study revealed that about half of the study subjects (50.3%, n = 148/300) took AstraZeneca, then the majority (24.5%, n = 72/300) took Moderna, and the rest of the respondents took Sino pharm(13.9%, n = 41/300)& Pfizer (33%, n = 11.2/300)as 1st & 2nd dose of vaccine. (Table 1).

### COVID-19 infection status among the health professionals

Among the all-health care professionals, nearly half (51%, n = 153/300) were revealed as COVID-19 test negative in contrast to 49% with test positive. (Fig 1).

### COVID-19 vaccination status among the health professionals

A greater part of health professionals (98%, n = 294/300) was found to be vaccinated, whereas a very negligible amount (2%, n = 6/300) was found yet none were vaccinated for different reasons. Among all the vaccinated health care professional's majority had received the 2nd dose (52.3%), more than one-third of the respondents took the 3rd dose (42.0%), and the very least number received only the 1st dose of vaccination (3.7%). A majority (68.7%) of the respondent had immediate health consequences after vaccination which includes pain in the vaccine site (39.1%), fever (26.0%), headache (13.7%), weakness (10.9%), joint pain (5.8%), nausea with vomiting (3.7%) and low blood pressure (0.7%). (Fig 2).

### Respondent's characteristics associated with the COVID-19 infection and immediate health effects after vaccination

Results of multivariate (cross table) analysis revealed that respondents' age (P<0.01), their type of organization (P = 0.05), and geographical location (P = 0.05) were found to be significant factors associated with positive COVID-19 infection status. On the other hand, immediate health effects after vaccination status among the respondents were significantly (P<0.01) associated with demographic characteristics like gender, profession, and geographic location. Additionally, the most important finding was, the brand of vaccine that a health professional received was significantly associated with positive infection status (P = 0.03) and immediate health effects after vaccination (P = 0.01). The study also revealed that the middle age group (30–39 years) was found to be more infected with COVID-19 (20.3%) where the comparatively highest infection rate was among female respondents (35%) who were nurses by their occupation (23%) while physicians were identified as a second largest group (13.7%) with highest

**Table 1. Characteristics of the respondents according to COVID-19 infection status and immediate health effects after vaccination (n = 300).**

| Characteristics | COVID-19 infection status | | | | Immediate health effects after COVID-19 vaccination | | | |
|---|---|---|---|---|---|---|---|---|
| | Number of participants, n (%) | Test Positive, n (%) | Test negative, n (%) | p-value (≤0.05) | Number of participants, n (%) | Yes, n (%) | No/NA, n (%) | p-value (≤0.05) |
| **Age group (in years)** | | | | | | | | |
| ≤29 | 83 (27.7) | 30 (10) | 53 (17.7) | 0.01* | 83 (27.7) | 57 (19) | 26 (8.7) | 0.48 |
| 30–39 | 115 (38.3) | 61 (20.3) | 54 (18.0) | | 115 (38.3) | 81 (27) | 34 (11.3) | |
| 40–49 | 75 (25) | 37 (12.3) | 38 (12.7) | | 75 (25) | 53 (17.7) | 22 (7.3) | |
| >50 | 27 (9) | 19 (6.3) | 8 (2.7) | | 27 (9) | 15 (5.0) | 12 (4.0) | |
| **Gender** | | | | | | | | |
| Male | 93 (31.0) | 42 (14.0) | 51 (17) | 0.37 | 93 (31.0) | 49 (16.3) | 44 (14.7) | 0.01* |
| Female | 207 (69.0) | 105 (35) | 102 (34) | | 207 (69.0) | 157 (52.3) | 50 (16.7) | |
| **Profession** | | | | | | | | |
| Physician | 85 (28.7) | 41 (13.7) | 45 (15) | 0.73 | 85 (28.7) | 59 (19.7) | 27 (9.0) | 0.01* |
| Nurse | 134 (44.7) | 69 (23) | 65 (21.7) | | 134 (44.7) | 105 (35.0) | 29 (9.7) | |
| Allied Health Professionals | 80 (26.7) | 37 (12.3) | 43 (14.3) | | 80 (26.7) | 42 (14.0) | 38 (12.7) | |
| **Type of Organization** | | | | | | | | |
| Private | 89 (29.7) | 36 (12) | 53 (17.7) | 0.05* | 89 (29.7) | 56 (18.7) | 33 (11) | 0.16 |
| Public | 211 (70.3) | 111 (37.0) | 100 (33.3) | | 211 (70.3) | 150 (50) | 61 (20.3) | |
| **Geographic Location** | | | | | | | | |
| Capital | 150 (50) | 65 (21.7) | 85 (28.3) | 0.05* | 150 (50) | 119 (39.7) | 31 (10.3) | 0.01* |
| Outside-capital | 150 (50) | 82 (27.3) | 68 (22.7) | | 150 (50) | 87 (29.0) | 63 (21.0) | |
| **Brand of vaccine taken in 1st & 2nd dose** | | | | | | | | |
| Pfizer | 33 (11.2) | 23 (7.8) | 10 (3.4) | 0.03* | 33 (11.2) | 21 (7.1) | 12 (4.1) | 0.01* |
| Moderna | 72 (24.5) | 34 (11.6) | 38 (12.9) | | 72 (24.5) | 56 (19.0) | 16 (5.4) | |
| Astrazenica | 148 (50.3) | 73 (24.8) | 75 (25.5) | | 148 (50.3) | 116 (39.5) | 32 (10.9) | |
| Sinopharm | 41 (13.9) | 14 (4.8) | 27 (9.2) | | 41 (13.9) | 13 (4.4) | 28 (9.5) | |

Data are presented as frequency (n), percentage (%);

*Statistical significance at p value ≤0.05.

Chi-square test was used to observe the association, NA = Not Applicable.

infection rate. Additionally, healthcare professionals who were residing outside of the capital city reported more COVID-19 infection (27.3%) compared to those from the capital city (21.7%). On the other hand, female subjects (52.3%) who were physicians (28.7%) and nurses (35.0%) as professionals experienced more vaccine-related immediate health effects compared to the other groups. Additionally, it was also observed that health professionals working in public organizations (50%) and in the capital city (39.7%) suffered from more immediate health effects compared to other study subjects. (Table 1).

## Predictors associated with COVID-19 infection and immediate health effects after vaccination among the respondents

Regression analysis of the study revealed significant predictors associated with COVID-19 infection and immediate health effects after vaccination among the respondents. The young adult group (30-39years: AOR = 2.01/0.03; CI: 1.08–3.76) and senior adult group (>50 years: AOR = 4.36/0.01; CI: 1.65–11.55) of all respondents found significantly having more positive infection compared to the younger and middle age groups (≤29 and 40–49 years of age). In addition, respondents from outside Dhaka (COR/P = 1.58/0.05; CI: 1.00–2.49) who were

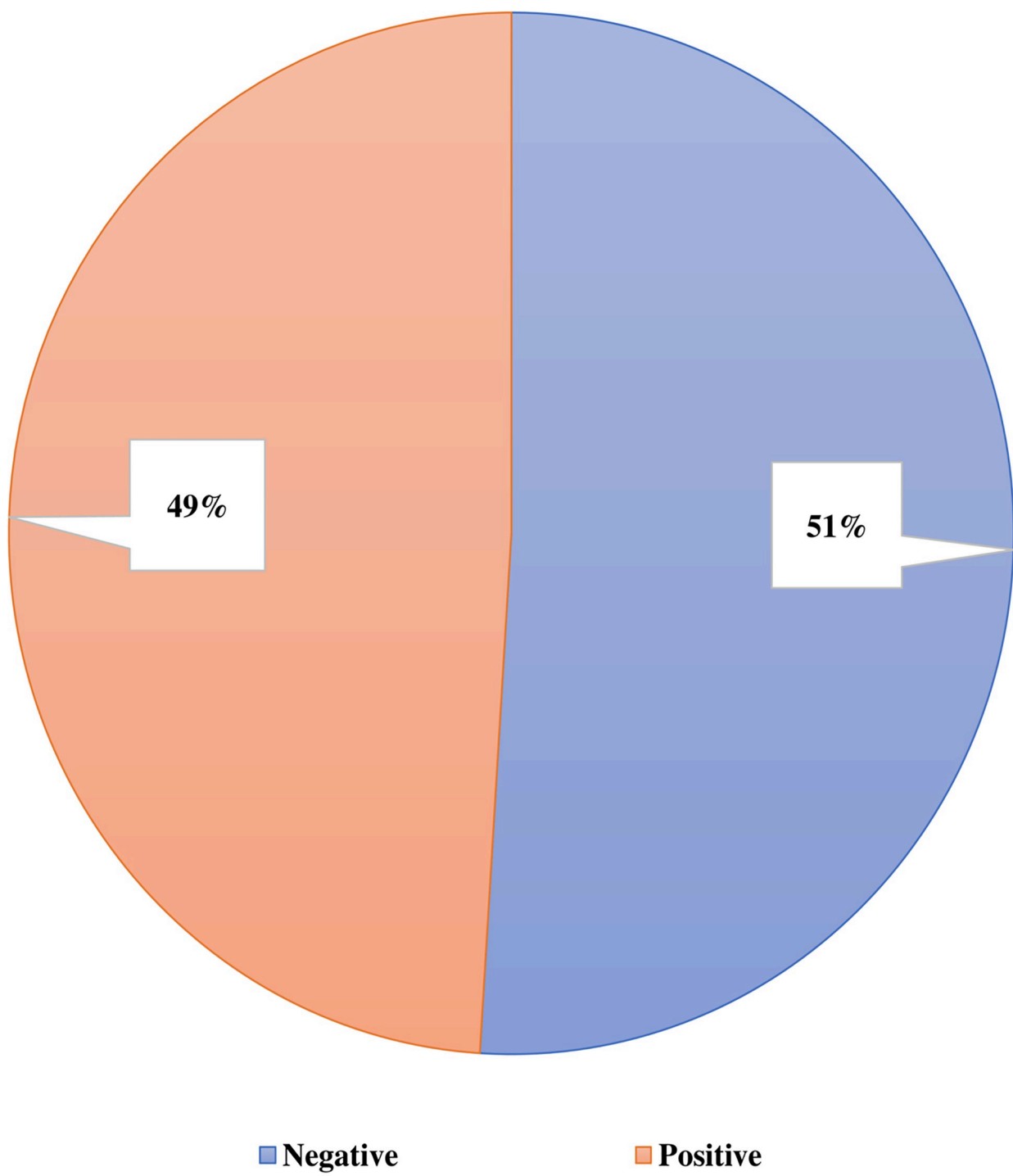

**Fig 1. This is the status of COVID-19 infection among the respondents (n = 300).**

working in private organizations (COR/P = 1.58/0.05; CI: 1.00–2.49) found moreCOVID-19 infected compared to the counter groups. Surprisingly, it is also significantly revealed that respondents who had COVID-19 infection history got more Sinopharm to correspond to Pfizer, Moderna, and Astra Zeneca as their 1st and 2ndvaccine dose. After the development of

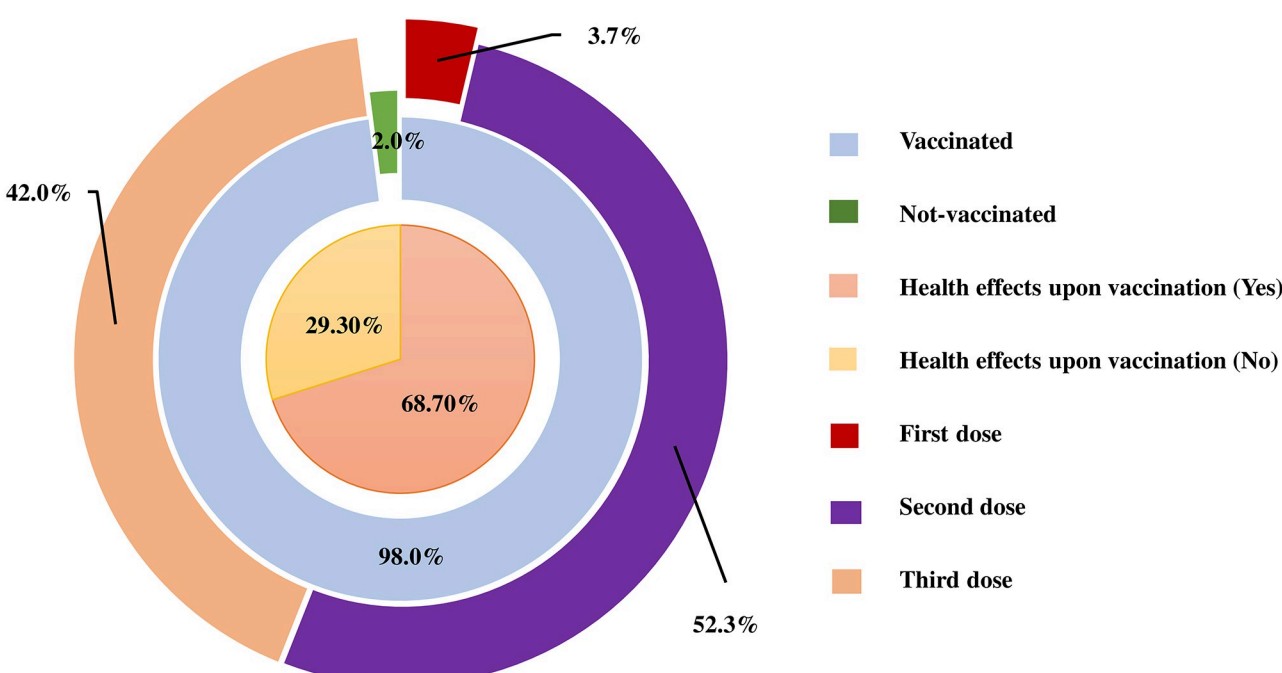

**Fig 2. This is the status of COVID-19 vaccination including immediate health consequences among the respondents (n = 300).**

the adjusted model along with the unadjusted predictors, possible confounders have been excluded by performing a backward elimination procedure with the precision of a 95% Confidence Interval. Therefore, final predictors for the COVID-19 infection status have been revealed as the young adult group (30–39 years), the senior adult group (>50 years), and the Sinopharm vaccine taken in 1st & 2nd doses. Significant predictors regarding experiencing immediate health effects after COVID-19 vaccination were revealed in this study. Interestingly at young aged (≤29 years: AOR = 3.23/0.03; CI:1.09–9.59), female gender (AOR = 3.31/0.01; CI:1.82–6.04), physicians (COR/P = 1.98/0.04; CI: 1.05–3.72) and nurses(COR/P = 3.28/0.01; CI: 1.79–5.98) as a profession who were working in the capital city Dhaka (AOR = 1.91/0.03; CI: 1.06–3.46) found as significant predictors to have more health effects after vaccination compared to the other groups. Furthermore, this study significantly reflected that respondents who received AstraZeneca as their vaccine were found to have less immediate health consequences on vaccination in comparison to other groups. Final predictors from the adjusted model after elimination of the confounders have been found as the young aged group (≤29 years), female as gender, respondents residing in the capital city Dhaka, and AstraZeneca as the vaccine for the 1st and 2nd dose. (Table 2).

## Discussions

We recruited 300 healthcare workers nationwide, where 50% (n = 150) were from different health facilities in the capital city and 50% from health facilities outside the capital city from other divisions. We believe this study will shed light to identify health workers' COVID-19 vaccination status with its immediate consequences and the predictor for immediate health consequences in Bangladesh. And this study revealed a greater coverage of vaccination with different immediate health consequences among the health professionals. However, still there is a lack of vaccination coverage with complete doses up to 3rd dose of vaccination.

**Table 2. Predictors associated with the COVID-19 infection status and side effects of vaccination among the respondents (n = 300).**

| Characteristics | COVID-19 infection status | | | | Immediate health effects after COVID-19 vaccination | | | |
|---|---|---|---|---|---|---|---|---|
| | Test positive vs test negative | | | | Yes, vs No/NA effect | | | |
| | Un-adjusted OR (95% CI) | P-value | Adjusted OR (95% CI) | P-value | Un-adjusted OR (95% CI) | P-value | Adjusted OR (95% CI) | P-value |
| **Age group (in years)** | | | | | | | | |
| ≤29 | 4.2 (1.6–10.7) | 0.01* | 4.2 (1.6–10.7) | 0.01* | 0.57 (0.23–1.39) | 0.22 | — | — |
| 30–39 | 2.1 (0.8–5.2) | 0.11 | 2.2 (0.9–5.6) | 0.08 | 0.53 (0.22–1.24) | 0.14 | — | — |
| 40–49 | 2.4 (0.9–6.3) | 0.06 | 2.5 (0.9–6.4) | 0.06 | 0.52 (0.21–1.29) | 0.16 | — | — |
| >50 | Reference | | | | Reference | | | |
| **Gender** | | | | | | | | |
| Male | 1.25 (0.76–2.04) | 0.37 | — | — | 2.8 (1.7–4.7) | 0.01* | 3.4 (1.9–6.1) | 0.01* |
| Female | Reference | | | | Reference | | | |
| **Profession** | | | | | | | | |
| Physician | Reference | | | | Reference | | | |
| Nurse | 0.86 (0.49–1.48) | 0.58 | — | — | 0.6 (0.3–1.1) | 0.11 | — | — |
| Allied Health Professionals | 1.06 (0.58–1.95) | 0.85 | — | — | 1.9 (1.1–3.7) | 0.03* | — | — |
| **Type of Organization** | | | | | | | | |
| Private | 1.6 (0.9–2.7) | 0.05* | — | — | Reference | | | |
| Public | Reference | | | | 1.45 (0.86–2.45) | 0.17 | — | — |
| **Geographic Location** | | | | | | | | |
| Inside Dhaka | 1.6 (1–2.5) | 0.05* | — | — | Reference | | | |
| Outside Dhaka | Reference | | | | 2.8 (1.7–4.6) | 0.01* | 2.1 (1.2–3.7) | 0.01* |
| **Brand of vaccine taken in 1st & 2nd Dose** | | | | | | | | |
| Pfizer | 2.6 (1.1–6.2) | 0.03* | 2.5 (1.0–6.1) | 0.05* | 0.5 (0.2–1.2) | 0.13 | 0.5 (0.2–1.4) | 0.19 |
| Moderna | 2.4 (1.1–5.3) | 0.04* | 2.6 (1.2–6.1) | 0.02* | 0.5 (0.2–1.1) | 0.08 | 0.5 (0.2–1.1) | 0.06 |
| Astrazenica | 4.4 (1.7–11.9) | 0.01* | 3.8 (1.4–10.5) | 0.01* | 3.8 (1.4–9.9) | 0.01* | 4.1 (1.5–11.3) | 0.01* |
| Sinopharm | Reference | | | | Reference | | | |

Logistic Regression Analysis was used to identify the predictors;

* Statistical significance at p value ≤0.05;

NA = Not Applicable, reference category was considered for COVID-19 infection status as test negative and for side effects of vaccination as No/NA.

Our study has revealed that dominant HCWs were found as female (69%) nurses (44.7%) and nearly half of the respondents (38.3%, n = 115/300) belonged to the age group of 30–39 years. Findings from a study in Congo show that (50.9%, n = 312/613) of the participants were men among a sample of 613 HCWs. The geographical location along with cultural variances & group of study participants stated the opposite findings [24].

Mostly they were from Government organizations and 49% were found as having COVID-19-positive infection. Around 10% of the total infection was found among health workers; in the early phase of this pandemic in Bangladesh [25]. These frontline fighters are additionally confronting huge challenges, including psychological suffering, and furthermore, they are assaulted by the negative impact of society [13]. Our study findings regarding infection status clearly represent the deficiency of health safety strategies for health care professionals.

As the Government of Bangladesh (GoB) launched a mass vaccination program in February 2021 with a prioritization approach for the health safety of frontiers like HCWs, the study found a greater part of health professionals (98%) as vaccinated. Among them, the highest portion already received 2nd dose of the COVID-19 vaccine. Our previous study showed that a

good number (18.3%) of HCWs were not vaccinated till August 2021 which indicates the advancement of the mass vaccination program in this current scenario [12]. United States weekly report shows that the groups with the highest percentage of reported fully vaccinated where health care professionals (75.1%) [26]. Study findings from the different areas of the world agree that vaccination status among the health care professionals in Bangladesh is still not satisfactory as mostly they don't have completed full vaccination package (1st, 2nd & 3rd dose). In this battle, all the doctors, nurses, pharmacists, health workers, law enforcement agencies, and others who have been fighting against COVID-19 nationally as well as globally should be vaccinated with all the doses.

The study significantly found young and senior adult (P<0.01) health care professionals from private organizations (P = 0.05) who were working throughout the country outside the Dhaka division (P = 0.05) found to have more positive COVID-19 infection rather than counter groups. It might be due to having comparatively poor knowledge of COVID-19 among the nurses which was reported in a study conducted in a nursing institute in Dhaka city Bangladesh [27]. A contrasting scenario was found in a study, where younger HCWs were found less likely to be infected by COVID-19 (P = 0.08) whereas the older age group was found to be more infected (P≤.0.01) [28].

On the other hand, adverse health effects of vaccination status were found more significant among the female (<0.01) respondents who were mostly nurses (<0.01) and physicians as professionals, working inside the capital (P<0.01) and took AstraZeneca vaccine as their 1st and 2nd dose. A study among the general population conducted in April 2021 showed that almost half got vaccinated till then and 57.41% developed immediate health consequences like fever, muscle pain, headache, and pain on the vaccination sites [18]. A study in Southern Ethiopia revealed that perception regarding side effects of vaccination among HCWs was significantly associated with their educational status while there was no influence on the age and occupational status of the respondents [29].

Concerning the predictors, the study found that the young adult (30–39 years) and senior adult (>50 years) groups of respondents found more significant to have positive infection rather than the middle and younger age (≤29 and 40–49 years of age) group of this study. A study identifying the highest infection and deaths reported among HCWs aged over 70 years supports our predictor regarding age although having different demographic settings [30]. Experiencing side effects due to COVID-19 vaccination young female respondents inside the capital city found a significant predictor. Furthermore, COVID-19 infected group were more significantly found to receive Sinopharm as their vaccine and fewer side effect was also observed among the significant group who received AstraZeneca as their 1st and 2nd dose. The study findings contradict another study of Israel regarding more side effects among males outside of the capital city [31].

In the war of massive COVID-19 infection rates, health care professionals are likely to work for prolonged periods under substantial pressures, along with the infection risk. As a consequence, the HCWs become steadily reluctant to do their work and psychologically weaken. With the limited healthcare facilities; hence, the healthcare professional's safety measures are a great concern with actions to reduce infection rate too. One of the key strengths of this study is that significant numbers of predictors triggering infection & vaccination status of the health care professionals will be helpful to draw further action plans by the scientific community. There are very few studies available indicating the COVID-19 infection and vaccination status combinedly among healthcare professionals from throughout Bangladesh incorporating authentic scientific procedures based on the COVID-19 infection rate in all 8 divisions. Thus, as another strength, the crucial outcome of this study is valid to generalize among the whole HCW population in Bangladesh. However, due to avoidance of the risk of COVID-19

contamination, a web-based data collection method was adopted in this present study which is considered a limitation of this study. This study might be the model research for further large-scale studies in the future with the possible way outs of action plans incorporating the possible health safety strategies to handle future sudden pandemics like COVID-19 and safety gourd strategies for the HCWs which lacks in this study.

## Conclusions

This study found a satisfactory scenario for the COVID-19 vaccination status among health care professionals throughout the country. In half of the cases, respondents were found to be COVID-19 infected. Though the majority of the HCWs revealed vaccinated there is a lack of doses coverage of vaccination among HCWs in Bangladesh. Therefore, full vaccination coverage with completion of all the doses among the HCWs especially in COVID-19 dedicated hospitals is pivotal for their health safety in the current pandemic. Otherwise, health services would be in a threatening situation. The study also revealed different immediate health consequences that occurred after vaccination among nearly three-quarters of the study subjects as well as explored the predictors associated with it. The vital outcome of this study recommends our government plan for a sustainable safety policy as well as strategies for sustainable development of the health sector of our country which will be capable enough to combat future pandemics like COVID-19. Thus, as frontline fighters of such health-related emergencies, HCWs will get the health safety opportunities to work dedicatedly for the affected population.

## Supporting information

**S1 File.**
(SAV)

## Acknowledgments

We strongly acknowledge the study participants and the authority of the study place.

## Author Contributions

**Conceptualization:** Bilkis Banu, Nasrin Akter, Sujana Haque Chowdhury.

**Formal analysis:** Bilkis Banu, Nasrin Akter, Sujana Haque Chowdhury.

**Investigation:** Md. Tanzeerul Islam, Muhammad Zahangir, Shah Monir Hossain, Sarder Mahmud Hossain.

**Methodology:** Bilkis Banu.

**Supervision:** Bilkis Banu, Kazi Rakibul Islam, Shah Monir Hossain, Sarder Mahmud Hossain.

**Validation:** Bilkis Banu.

**Visualization:** Bilkis Banu, Md. Tanzeerul Islam.

**Writing – original draft:** Bilkis Banu, Nasrin Akter, Sujana Haque Chowdhury, Kazi Rakibul Islam, Md. Tanzeerul Islam, Muhammad Zahangir.

**Writing – review & editing:** Bilkis Banu, Nasrin Akter, Sujana Haque Chowdhury, Kazi Rakibul Islam.

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
