## [Decision Letter · Decision Letter 0]

12 Sep 2022

PONE-D-22-15479COVID-19 and Bangladeshi Health Professionals: Infection status, vaccination and its immediate health consequencePLOS ONE

Dear Dr. Banu,

Thank you for submitting your manuscript to PLOS ONE. After careful consideration, we feel that it has merit but does not fully meet PLOS ONE’s publication criteria as it currently stands. Therefore, we invite you to submit a revised version of the manuscript that addresses the points raised during the review process.

We look forward to receiving your revised manuscript.

Kind regards,

Raphael Mendonça Guimaraes, PhD

Academic Editor

PLOS ONE

Journal Requirements:

Reviewers' comments:

Reviewer's Responses to Questions

**Comments to the Author**

1. Is the manuscript technically sound, and do the data support the conclusions?

Reviewer #1: Yes

Reviewer #2: Yes

2. Has the statistical analysis been performed appropriately and rigorously? 

Reviewer #1: Yes

Reviewer #2: Yes

3. Have the authors made all data underlying the findings in their manuscript fully available?

Reviewer #1: Yes

Reviewer #2: Yes

4. Is the manuscript presented in an intelligible fashion and written in standard English?

Reviewer #1: No

Reviewer #2: Yes

5. Review Comments to the Author

Reviewer #1: This is an important research in the field of COVID-19. I have some specific comments about your manuscript

Abstract

Introduction: The justification should be more succinct as it is in the body of the literature. The objective should be explicitly described. Consider my suggested objective in the manuscript.

Method: You need to remove the sentence on the composition of the health workers. Rather, you should have the information on the content of the questionnaire in this section.

Results: You need to add the 95% CI for the infection rates.

Introduction

Line 85, please write out the full meaning of SAGE if its appearing here for the first time.

Line 103, what do you mean by ‘novel professional’? Please this should be clearer or choose the appropriate word.

Please add the objective of this study to the end of the introduction section.

The justification/rationale for this study within the introduction is too lengthy for an article. Please endeavor to summarize it.

Method

You had no information on the study setting; the readers may be interested in this. The prevention measures prevalent in the hospital, presence of a functional IPC team, nature of COVID-19 immunization activities, actions for AEFIs following Covid-19 vaccination etc. You can cite reference, if it has been previously published.

Mention the inclusion and exclusion criteria for the respondents.

Mention the sampling procedure for selecting the hospitals.

For the sample size, just quote your minimum sample size (372). You do not need to start explaining the non-response rate. It is a finding and should strictly be in the result section.

Line 135, p=0.41 from the reference cited, and not p=0.59, please replace.

How did you check for duplicates of entries since it was an online based questionnaire?

You need to report the source of the 10 respondents used for the pretest.

It is not clear how the infection status of the respondents was determined? Please explain or cite reference, if it has been previously published.

Results

Add your response rate (316/372) 84.9%.

Line 183; add the mean age of the respondents or as the case may be. Unless, it has been earlier published.

Line 229-234, report only the adjusted Odds Ratio and its 95% CIs.

Discussion

Line 255-264, the first paragraph of the discussion should start with stating what the aim of the work was and a summary of your findings.

Your recommendation still sounds like the rationale for the study. Please rephrase appropriately based on your conclusion.

You had no statement on limitation, especially being an online-based questionnaire, likewise on the generalizability of your findings. Please can you explain this? Otherwise, address them in the body of the manuscript.

A lot of grammatical errors please address them. I was able to address some in the manuscript.

Reviewer #2: Considerations:Thank you for the opportunity to review this article.

Overall:The article is well written and substantiated. I congratulate the authors for their initiative.

Title: Well presented

Abstract:

The abstract is longer than 300 words

Introduction:

On line 89 the acronym SAGE was placed without describing what it means. The first time an acronym is recorded it should say what it means.

I would add in line 64, the number of active health workers in Bangladesh, because by putting only the number of deaths you don't have the dimension of how many died, if it was a lot or a little? This enriches the argument about how serious the number of deaths among health workers is.

In line 78 : "through then, we could eradicate small pox & nearly eliminate the wild-polio.[14]" Place the quote before the period.

Method:

A detailed description is needed, especially of the reliability and validity of the questionnaire.

Further detail the eligibility and exclusion criteria for participants.

How was the question of COVID-19 infection, was it considered what types of testing? How did you guys consider this data?

Results:

Better specify what Immediate health consequences of COVID-19 vaccination would be.

Discussion:

Well presented

Conclusion:

Resume in the conclusion what the immediate health consequences of having had an infection and being vaccinated are for the health of the health care worker. What happens if the professional is infected? And what are the repercussions of vaccination in this professional category?

This is the moment to value your findings.

6. PLOS authors have the option to publish the peer review history of their article (what does this mean?). If published, this will include your full peer review and any attached files.

Reviewer #1: No

Reviewer #2: No

---

## [Author Response · Author response to Decision Letter 0]

26 Sep 2022

Reviewer's Comment Response

Reviewer 1

Abstract

1. Introduction: The justification should be more succinct as it is in the body of the literature. The objective should be explicitly described. Consider my suggested objective in the manuscript. All indications are addressed in line numbers 22 to 28

We did not find any suggested objective. However, we modified it also in 26-28 lines.

2. Method: You need to remove the sentence on the composition of the health workers. Rather, you should have the information on the content of the questionnaire in this section. Removed and included information from questionnaire in line 32 and 33.

3. Results: You need to add the 95% CI for the infection rates. Added in line 38, 40, 43 & 44

Body of the manuscript 

4. Line 85, please write out the full meaning of SAGE if it appearing here for the first time. Added in line 99. 

5. Line 103, what do you mean by ‘novel professional’? Please this should be clearer or choose the appropriate word. Modified in line 68-71

6. Please add the objective of this study to the end of the introduction section.

The justification/rationale for this study within the introduction is too lengthy for an article. Please endeavor to summarize it. Objective already mentioned in line 118-121;

Justification is modified and concised.

7. You had no information on the study setting; the readers may be interested in this. The prevention measures prevalent in the hospital, presence of a functional IPC team, nature of COVID-19 immunization activities, actions for AEFIs following Covid-19 vaccination etc. You can cite reference, if it has been previously published.

Mention the inclusion and exclusion criteria for the respondents. Included in the manuscript in line 132, 135-138;

Inclusion and exclusion criteria are mentioned in line 160-162;

Mention the sampling procedure for selecting the hospitals. Mentioned in line 135-38 and 158-60

Line 135, p=0.41 from the reference cited, and not p=0.59, please replace. Modified in line 150

How did you check for duplicates of entries since it was an online based questionnaire?

You need to report the source of the 10 respondents used for the pretest. Mentioned in line 165-67;

Information regarding pretest mentioned in line 184-86;

It is not clear how the infection status of the respondents was determined? Please explain or cite reference, if it has been previously published. Included in line 189.

Results: Add your response rate (316/372) 84.9%. Added in line 206-07.

Line 183; add the mean age of the respondents or as the case may be. Unless, it has been earlier published. Added in 208.

Line 229-234, report only the adjusted Odds Ratio and its 95% CIs. Incorporated

Discussion: Line 255-264, the first paragraph of the discussion should start with stating what the aim of the work was and a summary of your findings. Incorporated in line 283-289

Your recommendation still sounds like the rationale for the study. Please rephrase appropriately based on your conclusion. Modified in the conclusion

You had no statement on limitation, especially being an online-based questionnaire, likewise on the generalizability of your findings. Please can you explain this? Otherwise, address them in the body of the manuscript. Added in line 351-53

A lot of grammatical errors please address them. I was able to address some in the manuscript. Already addressed

Reviewer 2

Abstract: The abstract is longer than 300 words Incorporated 

On line 89 the acronym SAGE was placed without describing what it means. The first time an acronym is recorded it should say what it means. Added in line 99.

I would add in line 64, the number of active health workers in Bangladesh, because by putting only the number of deaths you don't have the dimension of how many died, if it was a lot or a little? This enriches the argument about how serious the number of deaths among health workers is. Incorporated in line 68-71.

In line 78: "through then, we could eradicate small pox & nearly eliminate the wild-polio.[14]" Place the quote before the period. Modified in line 86-88.

A detailed description is needed, especially of the reliability and validity of the questionnaire.

 Mentioned in line 84.

Further detail the eligibility and exclusion criteria for participants. Added in line 160-62

How was the question of COVID-19 infection, was it considered what types of testing? Yes, mentioned in line 189

How did you guys consider this data? Quantitative 

Results: Better specify what Immediate health consequences of COVID-19 vaccination would be. Modified and included in line 227-230.

Resume in the conclusion what the immediate health consequences of having had an infection and being vaccinated are for the health of the health care worker. What happens if the professional is infected? And what are the repercussions of vaccination in this professional category? Incorporated in the conclusion

Editor's Comment

Minor grammatical errors throughout the manuscript require correction. Already addressed

---

## [Editor Report · Decision Letter 1]

18 Oct 2022

COVID-19 and Bangladeshi Health Professionals: Infection status, vaccination and its immediate health consequences

PONE-D-22-15479R1

Dear Dr. Banu

We’re pleased to inform you that your manuscript has been judged scientifically suitable for publication and will be formally accepted for publication once it meets all outstanding technical requirements.

Kind regards,

Raphael Mendonça Guimaraes, PhD

Academic Editor

PLOS ONE
---

## [Editor Report · Acceptance letter]

21 Oct 2022

PONE-D-22-15479R1 

COVID-19 and Bangladeshi Health Professionals: Infection status, vaccination and its immediate health consequences 

Dear Dr. Banu:

I'm pleased to inform you that your manuscript has been deemed suitable for publication in PLOS ONE. Congratulations! Your manuscript is now with our production department. 

Kind regards, 

on behalf of

Dr. Raphael Mendonça Guimaraes 

Academic Editor

PLOS ONE